

# A bilingual benchmark for evaluating large language models

Mohamed Alkaoud

Department of Computer Science, College of Computer and Information Sciences, King Saud University, Riyadh, Saudi Arabia

## ABSTRACT

This work introduces a new benchmark for the bilingual evaluation of large language models (LLMs) in English and Arabic. While LLMs have transformed various fields, their evaluation in Arabic remains limited. This work addresses this gap by proposing a novel evaluation method for LLMs in both Arabic and English, allowing for a direct comparison between the performance of the two languages. We build a new evaluation dataset based on the General Aptitude Test (GAT), a standardized test widely used for university admissions in the Arab world, that we utilize to measure the linguistic capabilities of LLMs. We conduct several experiments to examine the linguistic capabilities of ChatGPT and quantify how much better it is at English than Arabic. We also examine the effect of changing task descriptions from Arabic to English and vice-versa. In addition to that, we find that fastText can surpass ChatGPT in finding Arabic word analogies. We conclude by showing that GPT-4 Arabic linguistic capabilities are much better than ChatGPT's Arabic capabilities and are close to ChatGPT's English capabilities.

## INTRODUCTION

In recent years, the development and utilization of advanced attention-based (*Vaswani et al., 2017*; *Bahdanau, Cho & Bengio, 2015*) language models have revolutionized not only natural language processing (NLP) but many other fields. These language models, often referred to as large language models (LLMs), are capable of processing and generating human-like text by learning patterns and structures from vast amounts of training data. While LLMs have gained significant attention and have been extensively evaluated for English (*OpenAI, 2023*; *Bubeck et al., 2023*; *Anil et al., 2023*; *Nori et al., 2023*; *Katz et al., 2023*), their application and evaluation in the context of non-English languages such as Arabic have received relatively less attention.

The importance of non-English evaluation methods for large language models cannot be overstated. While English is one of the most widely spoken languages in the world, it is essential to recognize and cater to the linguistic diversity present across different cultures and regions. By solely relying on English evaluation methods, we risk neglecting the needs and perspectives of non-English speakers, limiting the applicability and effectiveness of

Corresponding author
Mohamed Alkaoud,
malkaoud@ksu.edu.sa

large language models in addressing global issues. For instance, Arabic, as one of the world's major languages, holds great significance in various domains, including communication, media, literature, politics, and religion. Yet, it receives less attention from researchers compared to English in NLP in general and specifically when dealing with LLMs.

The problem that we noticed—in both academia and industry—is that there is no clarity regarding the best LLM in Arabic. Some researchers decide to use to ChatGPT for Arabic since it is the most widespread; others decide to pay more and use GPT-4 because it is generally better in English and they think that will extend to Arabic. Some companies choose to pursue open source LLMs for their Arabic tasks; while others decide that training a specific Arabic LLM is the way to go. The main issue is that evaluating LLMs performance is in itself a difficult topic. Hence, knowing the answers to these questions is not clear. Two of the main questions concerning LLMs that we have seen from the Arabic NLP community are the following:

1. How well does an LLM perform in Arabic?
2. Does an LLM $x$ have similar Arabic capabilities as English?

When dealing with the evaluation of non-English languages, we face two different issues: (1) the need for a benchmark that evaluates LLMs in a given language, and (2) the need for a bilingual benchmark which results that can be compared across languages. To elaborate more on the second point, imagine an English benchmark $b_1$ giving some LLM $L_1$ a score of 71%. Now, imagine an Arabic benchmark $b_2$ giving $L_1$ a score of 50%. Can we say with high certainty that $L_1$ is better in English than Arabic? The answer is no; since we cannot compare the scores of two different tests ($b_1$ and $b_2$) directly.

To highlight the importance of the second question it is important to remember that while the main goal of English LLMs today is to improve performance further, in other languages such as Arabic, the goal can simply be matching the performance of English. We notice this trend in other languages and tasks where researchers aim to reach the level of performance in English—being the most researched and developed NLP language. In this work, we propose a novel evaluation method in both Arabic and English with results that are comparable between the two languages.

To accomplish our objective, we take one of the leading standardized tests in the middle east: the General Aptitude Test (GAT) (*NCA, 2023a*). GAT, similar to the Scholastic Aptitude Test (SAT) required by most American colleges and universities, is a standardized test required by many universities in Saudi Arabia and the Arab world. The General Aptitude Test is developed and managed by the National Center for Assessment (NCA) (*NCA, 2023b*). The exam is of the utmost importance to students since the results of the GAT exam play a significant role in determining students' eligibility for admission into higher education institutions. Universities in Saudi Arabia often require students to achieve a certain minimum score on the GAT exam to qualify for admission. Additionally, the exam results may also influence scholarship opportunities and the selection process for specialized academic programs. As such, the GAT exam holds great importance for students as they strive to secure their educational aspirations. For instance, King Fahd University of Petroleum and Minerals (KFUPM) (*KFUPM, 2023*), one of the most desired universities in Saudi Arabia with an acceptance rate of 4% (*KFUPM, 2023*), bases its admission on a

compound score out of Half of that score is determined by the student's performance on the GAT exam.

The GAT was introduced in 2003 to serve as a crucial tool in the selection process for university admissions in the Kingdom and other Arab nations. Initially, the exam was only offered in Arabic; however, due to the growth of the education sector and the emergence of numerous private higher learning institutions that use English as their main language of instruction, there was a need for an English version of the test to accommodate non-native Arabic speakers. According to the NCA, The GAT English Version is not a mere translation of its Arabic counterpart (*National Center for Assessment, 2023a*) but an independent test specifically designed for English speakers. As such, it ensures fair assessment standards for individuals taking the GAT, regardless of whether they opt for the Arabic or English version. The NCA guidelines specify that there is no difference between the scores of the Arabic and English tests and that they can be compared directly.

In this work, we use the GAT exam as a method to evaluate the performance of LLMs in both Arabic and English. While it is very difficult to come up with a bilingual test that has results that can be compared across languages, the NCA spent decades carefully designing such an exam. The findings of this research will not only contribute to the current body of knowledge in Arabic NLP but also provide practical insights for researchers, practitioners, and developers working on Arabic (and other non-English) language-related applications. Our contributions can be summarized as:

1. We propose a new way of evaluating LLMs in Arabic and English.
2. We build the first GAT questions dataset which can be used to evaluate four linguistic tasks: reading comprehension, word analogies, contextual errors, and sentence completion.
3. We build the first bilingual (Arabic-English) questions dataset with results that can be compared directly between the two languages.
4. We show that traditional word embeddings (fastText) can surpass the performance of ChatGPT in finding word analogies in Arabic.
5. We study the effect of alternating the instructions given to ChatGPT from Arabic to English and vice-versa.
6. We show that GPT-4's Arabic capabilities are substantially better than ChatGPT.

The rest of the article is structured as follows: in 'Background', we give a concise overview of Arabic NLP and the GAT exam; in 'Related Work', an overview of previous efforts in evaluating LLMs is discussed; and in 'Data Collection', the process of collecting and gathering the questions for our benchmark is detailed. 'Approach' explains the process of using our benchmark to evaluate an LLM and the details of using fastText to answer word analogy questions. 'Results and Discussion' dives into the results of our benchmark on ChatGPT and GPT-4. Finally, in 'Conclusion', we conclude the article with a brief summary.

## BACKGROUND

Arabic, as one of the world's major languages, holds significant cultural, historical, and linguistic importance. It plays a vital role in connecting communities and fostering cultural

exchange. It serves as the language of the Qur'an, the holy book of Islam, and is deeply intertwined with Islamic civilization and scholarship. With its complex grammar and its highly inflected nature, Arabic challenges computational linguists and NLP researchers, prompting further study of its linguistic intricacies computationally. This has led to a trend of building Arabic-specific transformer-based NLP language models because it has been shown that language-specific transformer-based models work better than generic multilingual ones (*Virtanen et al., 2019*; *Antoun, Baly & Hajj, 2020*; *de Vries et al., 2019*; *Martin et al., 2020*) which led to the development of Arabic-specific BERT (*Devlin et al., 2019*) models (*Antoun, Baly & Hajj, 2020*; *Abdul-Mageed, Elmadany & Nagoudi, 2021*; *Inoue et al., 2021*), an Arabic-specific ELECTRA model (*Clark et al., 2020*; *Antoun, Baly & Hajj, 2021a*), an Arabic-specific T5 model (*Raffel et al., 2020*; *Nagoudi, Elmadany & Abdul-Mageed, 2022*), an Arabic-specific BART model (*Lewis et al., 2020*; *Kamal Eddine et al., 2022*), and an Arabic-specific GPT2 model (*Radford et al., 2019*; *Antoun, Baly & Hajj, 2021b*). While these models do certainly show better results in Arabic compared to the original models, we believe that this trend will no longer continue with LLMs due to their enormous training costs that are out-of-reach for most if not all Arabic NLP researchers; for instance, *Touvron et al. (2023)* mention in their paper that they needed to run 2048 NVIDIA A100 GPU with 80GB of RAM for approximately 21 days to train the LLaMA 65B model. This makes the Arabic-specific evaluation of LLMs more important since it seems moving forward most Arabic-supporting LLMs will be multilingual and not language-specific.

The GAT is a standardized test introduced in 2003 that is widely recognized and used for college admissions in the middle east. Developed and administered by the NCA, the GAT assesses a student's readiness for college-level academics by evaluating their skills in language understanding and mathematics. The test is typically taken by high school students during their junior or senior year, and GAT scores are considered by colleges and universities as part of their admission criteria. The GAT consists of multiple-choice questions spanning two sections: verbal and quantitative. The verbal section measures a student's ability to comprehend and analyze written passages, understand and contrast analogies, find out-of-context words, and examine the best ways to construct sentences. The quantitative section assesses a student's problem-solving abilities and understanding of mathematical concepts, ranging from basic arithmetic to algebra, geometry, and data analysis. The GAT is designed to measure a student's aptitude and readiness for higher education, providing colleges with a standardized metric to compare and evaluate applicants from diverse backgrounds. GAT scores play a crucial role in the college admissions process, serving as one of the factors considered by institutions when reviewing applications. Alongside other application components such as high school grades, GAT scores help colleges assess a student's academic potential and their ability to succeed in college-level coursework. Additionally, the GAT offers students the opportunity to showcase their skills and academic achievements, allowing them to demonstrate their preparedness for higher education and potentially gain admission to their desired colleges or universities.

# RELATED WORK

While there have been many works relating to measuring how good LLMs can perform on specific exams, due to the novelty of our task within the NLP community, there is currently no available dataset that completely aligns with our specific requirements: (1) measuring linguistic capabilities in both Arabic and English, and (2) enabling the direct comparisons of these measures across two languages. *Hendrycks et al. (2021b)* and *Hendrycks et al. (2021a)* introduced the MMLU benchmark, a set of English multiple-choice questions spanning 57 fields. *OpenAI (2023)* translated these questions into 26 languages using machine translation (Microsoft Azure Translate) and measured GPT-4 performance on the original and translated versions. *Clark et al. (2018)* proposed a grade-school question dataset named AI2 Reasoning Challenge (ARC) that contains mostly science questions. One unique characteristic about ARC is that they are split into two groups based on difficulty: challenge and easy. DROP (*Dua et al., 2019*) (Discrete Reasoning Over the content of Paragraphs) is a large reading comprehension benchmark introduced by *Dua et al. (2019)* that contains over 96,000 questions extracted from Wikipedia. The questions focus on two domains: sports and history.*Cobbe et al. (2021)* proposed GSM8K, a set of around eight thousand grade-school-level math questions. The questions in GSM8K can be solved primarily using basic arithmetic operations (addition, subtraction, multiplication, and division). In Google's Pathways Language Model (PaLM) technical report, *Anil et al. (2023)* demonstrate Google's PaLM model performance on multiple language exams covering five languages: Chinese, Japanese, Italian, French, and Spanish. *Nori et al. (2023)* evaluated GPT-4 on the United States Medical Licensing Examination (USMLE): a three-step examination that physicians must pass in order to obtain a medical license to practice medicine in the United States that is sponsored by the Federation of State Medical Boards (FSMB) and the National Board of Medical Examiners (NBME). *Katz et al. (2023)* investigated the performance of GPT-4 on legal language, which is often sophisticated and complex, by giving it (GPT-4) questions from the bar exam.

There has been some work on investigating LLMs' multilingual behavior. *Lai et al. (2023)* conducted an evaluation of ChatGPT across seven distinct tasks (part-of-speech tagging, named entity recognition, relation extraction, natural language inference, question answering, common sense reasoning, and summarization) and 37 diverse languages. Their findings indicate that ChatGPT's performance is suboptimal for NLP tasks across various languages, emphasizing the need for task-specific models to ensure optimal performance. *Bang et al. (2023)* research on the multilingual evaluation of ChatGPT demonstrates that its understanding of non-Latin scripts (Chinese and Korean) is superior to its ability to generate them.

When it comes to Arabic-focused LLM evaluation, there is not much work in the area. We will discuss three related works and their limitations. First, in OpenAI's (*OpenAI, 2023*) GPT-4 report, the authors used machine translation to translate an English questions data set (MMLU (*Hendrycks et al., 2021b*; *Hendrycks et al., 2021a*)) to Arabic. While this can be thought of as a simple experiment to showcase GPT-4's performance in Arabic, it cannot be considered an accurate benchmark for the Arabic linguistic capabilities of an LLM. Second,

*Khondaker et al. (2023)* evaluated ChatGPT's performance on many NLG and NLU tasks spanning 60 datasets. The NLG tasks they use are machine translation, code-switching, summarization (and title generation), question answering (and generation), transliteration, paraphrasing, text rewriting, grammatical error correction, dialogue generation, and diacritization. The NLU tasks they use are emotion and sentiment analysis, dialect classification, claim and machine-generation detection, toxic text classification, irony and sarcasm classification, adult and dangerous content classification, demographic text classification, word sense disambiguation, and text-pair classification tasks. Third, *Abdelali et al. (2023)* evaluated LLM performance on many NLP (similar to *Khondaker et al., 2023*) and speech tasks. In the latter two works discussed, the main goal is to evaluate the performance of LLMs compared to other NLP models which leads us to answers to questions such as: should we use LLMs for sentiment classification instead of developing models specific to sentiment classification? In contrast, we aim to have a measure of how good are LLMs in Arabic compared to English (being the most developed language for NLP and LLMs) and propose a new dataset to help us in quantifying that.

## DATA COLLECTION

The GAT exam includes two sections: verbal and quantitative, as discussed in Section 'Background'. All questions in both sections are multiple choice with four choices for each question (A, B, C, and D). While the quantitative section is composed of questions that test basic arithmetic, algebra, geometry, and data analysis, the verbal section consists of four different styles of questions: (1) *reading comprehension*: you are given a passage, and then you have to answer questions related to the given passage; (2) *analogy*: you are provided with a pair of words that express a particular connection and then you are required to select the pair of words that most closely resembles the relationship in the original pair; (3) *contextual error*: you are given a sentence with four highlighted words, and then you have to select which word does not fit the meaning of the sentence; and (4) *sentence completion*: you are provided with a sentence or two with one or two missing words, and then you should choose the best word/words that best fit. Figure 1 shows an example of each question type in the verbal section of the GAT.

While the GAT includes two sections, we decided to only focus on the verbal section in this work due to the following reasons:

1. The verbal section focuses on linguistic abilities which we want to measure; on the other hand, the quantitative section focuses more on mathematics and quantitative capabilities.

2. Many of the questions in the quantitative section contain graphs, plots, or diagrams that can be tricky to include in an LLM. While there are multi-modal LLMs today that can deal with images in addition to text, the majority of LLMs today are text-based. Moreover, including images will make it difficult to explain if the results of a certain model are degraded because it failed to encode images accurately or because it failed to understand and solve the question correctly.

We collected the data from two main sources: (1) official questions published by the NCA, and (2) questions used on unofficial GAT guides. It was challenging for us to find

| Question type | English example | Arabic example |
|---|---|---|
| **Reading comprehension** | Read the following passage, then choose the best answer to each of the questions that follow and mark your choice.<br><br>(1) While there are more than a thousand different types of bananas around the world, most of us, sadly, only know the Cavendish variety. Cavendish bananas are hearty and can survive overseas shipping, but they are definitely not the most delicious.<br>(2) That title belongs to ice-cream bananas, which can be found easily in Hawaii, as well as other tropical environments in Southeast Asia and Central America. Like its name suggests, the ice-cream banana is sweet with undeniable hints of vanilla. At its most ripe stage, its texture is fluffy and creamy, and it practically melts in your mouth.<br><br>Which of the following is an opinion?<br><br>A) Ice-cream bananas can be found in tropical environments.<br>B) The most well-known variety of bananas is the Cavendish.<br>**C) Ice-cream bananas are the most delicious banana.**<br>D) Cavendish bananas travel well.vanilla. At its most ripe stage, its texture is fluffy and creamy, and it practically melts in your mouth. | الأسئلة التالية تتعلق بالنص الذي يسبقها، بعد كل سؤال أربعة اختيارات، أحدها صحيح. المطلوب هو: قراءة النص بعناية، واختيار الإجابة الصحيحة عن كل سؤال.<br><br>تختلف الأنهار فيما بينها اختلافاً كبيراً من حيث الحجم، فبعضها صغير جداً حتى إنها تجف خلال فصول الجفاف. وأطول نهر في العالم هو نهر النيل في إفريقيا، ويليه من حيث الطول نهر الأمازون في أمريكا الجنوبية، إلا أن كمية المياه التي يحملها نهر الأمازون تفوق كمية المياه في أي نهر آخر، بل وتفوق كمية المياه في نهر النيل، ونهر المسيسيبي في أمريكا، ونهر يانجستي في الصين، مجتمعة.<br>وإلى جانب كون الأنهار مهمة للزراعة، فهي مصدر مهم للطاقة؛ إذ يمكن استخدام قوة تدفق المياه على امتداد النهر، عند المساقط وغيرها من المناطق المنحدرة؛ لتشغيل الآلات، وتوليد الكهرباء، حيث تحوّل السواقي والدواليب المائية قوة المياه المتدفقة إلى طاقة. وكانت الطواحين والورش ومصانع النسيج تقام في الماضي بالقرب من الأنهار المنحدرة، وكانت تديرها قوة اندفاع المياه. وفي الوقت الحاضر، تنتج محطات القوة الكهرومائية، ذات التوربينات المائية، نحو ربع القوة الكهربائية التي يحتاجها العالم.<br><br>نهر النيل، من حيث طوله وكمية مائه، مقارنة بنهر الأمازون:<br><br>أ) أقصر منه، وأقل ماءً<br>ب) أطول منه، وأكثر ماءً<br>ج) أقصر منه، وأكثر ماءً<br>**د) أطول منه، وأقل ماءً** |
| **Analogy** | In the following questions there is a pair of capitalized words followed by four choices marked A, B, C, D. Choose the pair of words whose relationship is most similar to that expressed by the capitalized pair and mark your choice.<br><br>COLD : HOT<br><br>A) handsome : young<br>**B) beautiful : ugly**<br>C) summer : spring<br>D) complete : total | في بداية كل سؤال ممّا يأتي، كلمتان ترتبطان بعلاقة معينة، تتبعهما أربعة أزواج من الكلمات، أحدها ترتبط فيه الكلمتان بعلاقة مشابهة للعلاقة بين الكلمتين في بداية السؤال، هو: اختيار الإجابة الصحيحة.<br><br>أسد: شبل<br><br>أ) خروف: كبش<br>ب) حيوان: جمل<br>**ج) دجاجة: كتكوت**<br>د) نسر: صقر |
| **Contextual error** | One of the four highlighted words in each of the following passages does NOT fit the meaning of the text. Choose that word and mark your choice.<br><br>Two **causes** are available for trading; private and public. Individual **traders** tend to use the private option, where the **proprietor** will take some **responsibility** for the items.<br><br>**A) causes**<br>B) traders<br>C) proprietor<br>D) responsibility | في كل جملة مما يأتي أربع كلمات كل منها مكتوبة بخط غليظ. المطلوب هو: تحديد الكلمة التي لا يتفق معناها مع المعنى العام للجملة، ثم اختيار الحرف المقابل لها الإجابة. (الخطأ ليس إملائياً ولا نحوياً)<br><br>للتغير المناخي الكلمة **الفصل** في الصحراء؛ ففي أزمنة الوفرة **تنأى** البحيرات؛ **فتزدهر** التجمعات البشرية في **أرجائها**.<br><br>أ) الفصل<br>**ب) تنأى**<br>ج) فتزدهر<br>د) أرجائها |
| **Sentence completion** | Choose the best answer to complete the following sentences and mark your choice.<br><br>When I was _____, I always thought my classes were very difficult and feared that when I got older my _____ would be impossible.<br><br>A) a child / movement<br>B) abroad / past<br>C) wiser / mission<br>**D) younger / studies** | تلي كل جملة من الجُمل الآتية أربعة اختيارات، أحدها يُكمل الفراغ أو الفراغات في الجملة إكمالاً صحيحاً. المطلوب هو: اختيار الإجابة الصحيحة.<br><br>استُخدم مصطلح العولمة أساساً _____ بعض الأوجه الرئيسة للتحول الحديث في النشاط _____ العالمي.<br><br>أ) لتسوية – السياسي<br>**ب) لوصف – الاقتصادي**<br>ج) لتشبيه – الأدبي<br>د) لتفنيد – الثقافي |

**Figure 1 Examples of the four question types in the verbal section of the GAT exam in both English and Arabic.** The correct choice of each question is indicated in bold. The questions were taken from from the official GAT guides for English (*National Center for Assessment, 2023b*) and Arabic (*National Center for Assessment, 2023c*).

good quality questions; this was even harder for English, due to the scarcity of data available to it compared to its Arabic counterpart. Moreover, many of these questions were found in images, image-based PDFs, and websites requiring you to log in which made the task even more cumbersome. The data collection task was initially done by us. However, it was apparent after a while that it was going to take us very long to even collect a small number

of questions; for instance, one of the practice exams we collect can only be accessed by logging in to the official NCA website. The main issue was that only test takers can log in and it wasn't possible to create an account without registering for the test. We then decided to utilize crowd-sourcing to enable us to collect and clean questions in a way that is quick and can easily scale. We utilized Khamsat (https://khamsat.com/) (an Arabic crowd-sourcing platform) to help us with two tasks: (1) extracting the questions from documents, (2) formatting those questions and grouping them according to their types (reading comprehension, analogy, contextual error, or sentence completion).

One question that comes to mind after seeing Fig. 1 is how to encode contextual error questions since they contained bold text; we encode bold text by surrounding it with two asterisks. For instance, the following question:

> Two **causes** are available for trading; private and public. Individual **traders** tend to use the private option, where the **proprietor** will take some **responsibility** for the items.

will be encoded as:

> Two *causes* are available for trading; private and public. Individual *traders* tend to use the private option, where the *proprietor* will take some *responsibility* for the items.

After gathering the data, we noticed that some questions were repeated multiple times. The problem was that those duplicate questions were sometimes not identical syntax-wise but had small variations instead. Instead of going manually through every possible pair of questions which would've taken quite long, we developed a simple approach that allows us to quickly swift through these cases. We run every question through a BERT model and generate a document/sentence vector using the mean of the last hidden state vectors. After that, we get the most similar pair using cosine similarity as shown in Eq. (1):

$$\underset{(x,y)\in\{(q_i,q_j)\in Q^2|q_i\neq q_j\}}{\operatorname{argmax}} \frac{v(x)\cdot v(y)}{\|v(x)\|\,\|v(y)\|} \tag{1}$$

where $Q$ is the set containing all questions, and $v(\cdot)$ is a function that returns a vector given a string. Now we can sort every pair of questions according to how similar they are (from most similar to least similar) in a list. We then take the first $k$ pairs (we set $k$ to 50 in our case) and manually check if there are any duplicate questions in that batch. If some questions in that batch were flagged as duplicates, then we also check another batch (with the most similar pairs of questions) after removing the previous batch from the list. If no duplicates are found, then we stop. Figure 2 illustrates our process for detecting duplicates.

We noticed an interesting case with duplicates: sometimes two questions have similar text but the provided choices are different. For instance, we have two analogy questions that both ask you to indicate which pair of words share similar relationships to *PUNCTURE: TIRE*. While both questions ask you basically the same thing, the given answers are different: in the first question we have the following choices:

A) inflate: balloon

B) retract: statement

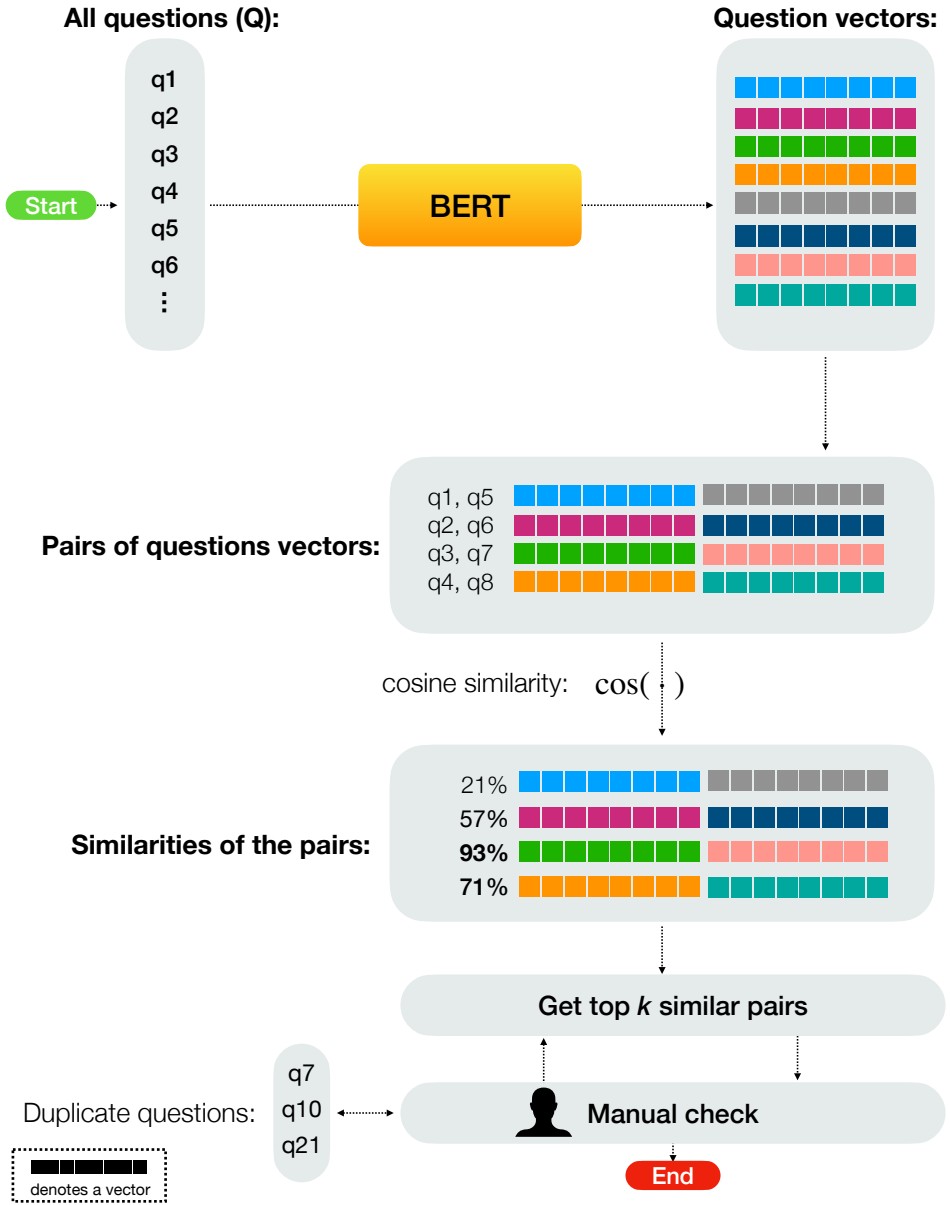

**Figure 2** Our questions' duplicate removal process: (1) we start by getting a vector representation (using BERT) for every question in our dataset, (2) we generate pairs of questions for all questions we have (we don't pair a question with itself), (3) we calculate the cosine similarity between each pair, (4) we get the top-$k$ most similar pairs and perform a manual check to see if there are any duplicate questions in the fetched pairs, (5) if we find at least one duplicate question, we remove one instance of the questions, and continue finding more duplicates by fetching the top-$k$ similar pairs again (after removing the previous top-$k$ similar pairs) and continue going back and forth, (6) finally, we stop if the fetched top-$k$ pairs do not contain any duplicates.

**Table 1 The number of questions in our dataset after removing duplicates.**

| Language | Question type | Number of questions |
|---|---|---|
| English | Reading comprehension | 91 |
| | Analogy | 124 |
| | Contextual error | 91 |
| | Sentence completion | 150 |
| | **Total** | **456** |
| Arabic | Reading comprehension | 140 |
| | Analogy | 121 |
| | Contextual error | 101 |
| | Sentence completion | 106 |
| | **Total** | **468** |

C) owe: favor

D) pierce: ear

while the second question contains these possible answers:

A) explore: curiosity

B) miser: poverty

C) gambler: winner

D) knight: beauty

We keep questions like these and do not remove them. Table 1 shows the number of questions collected for each language after removing duplicates.

## APPROACH

We start by constructing a prompt for every question in our dataset. To avoid biases and unscientific prompt engineering practices, the prompt we use is exactly the same as the one used in an official GAT exam (taken from the official GAT guides for Arabic (*National Center for Assessment, 2023b*) and English (*National Center for Assessment, 2018*) respectively). Another reason we use the official GAT question explanations is because that is how exam takers are tested and we did not want to unintentionally add our biases by designing our own prompts. We then take every question we have and construct a complete prompt for that question as shown in Fig. 3 After that, we use OpenAI's ChatGPT (gpt−3.5-turbo) API to ask the model to answer the question. The reason we use ChatGPT is that it is considered to be the golden standard when it comes to LLMs; no other model—sans GPT-4—has matched ChatGPT's performance (*Gudibande et al., 2023*). In addition to that, it is multilingual and supports Arabic in contrast to many LLMs available today. We fetch ChatGPT's response and use a hybrid approach to figure out the choice chosen by the LLM. The hybrid algorithm we use is highlighted in Algorithm 1. The algorithm will go over all possible choices $C$ (in our case $C$ will include the letters: A, B, C, and D; and their corresponding Arabic letters: أ, ب, ج and د ), and check if each

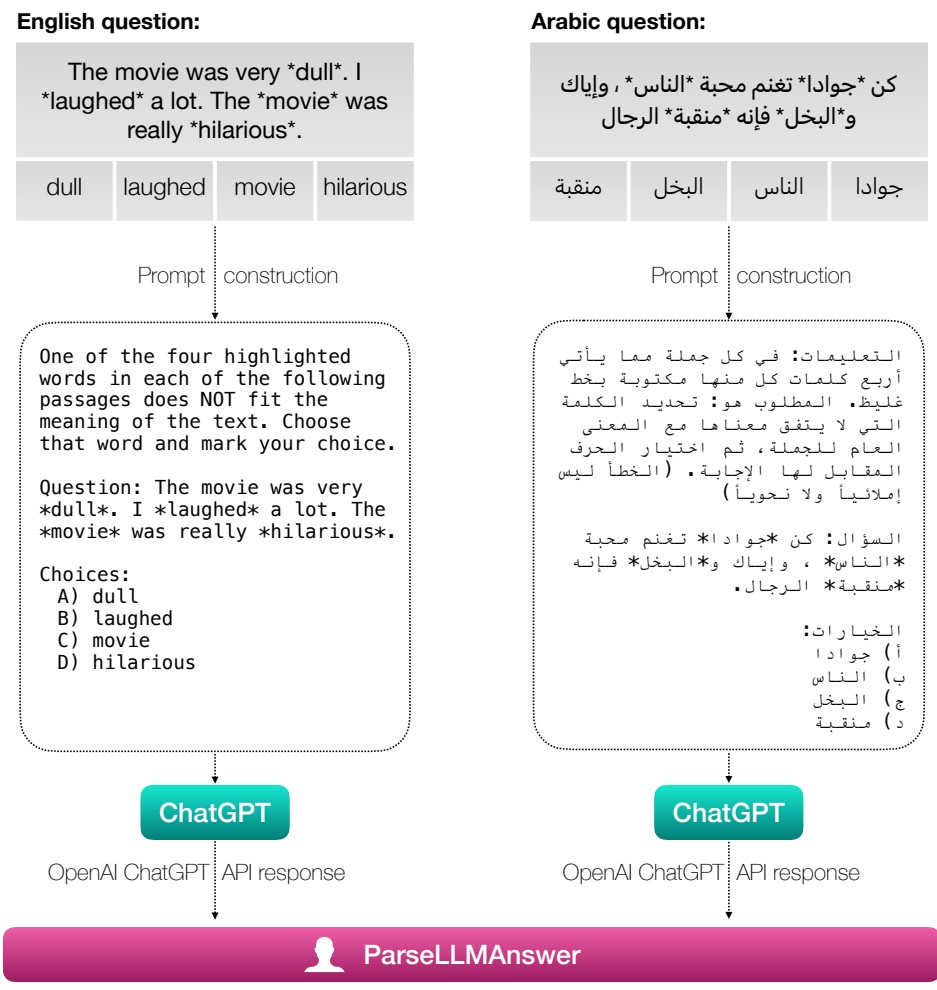

**Figure 3** The process of going from structured questions to LLM answers: (1) we create a prompt using the official GAT instructions for each question type for each language, (2) we send the prompt to the ChatGPT API, and (3) we use the ParseLLMAnswer algorithm (Algorithm ??) to extract the LLM's answer.

is in the LLM's response. If only one choice is found, then we consider it to be the LLM chosen answer. For other cases, where we have: (1) more than one choice, or (2) no choices returned; we conduct a manual check by investigating the response and mapping it to a choice. The reason this situation happens is that sometimes the LLM returns the correct answer without indicating which choice (A, B, C, or D) it is. In other cases, the LLM returns all choices with the correct choice highlighted (using Markdown).

The reason we check for both Arabic and English letters for all questions regardless of the language is that we noticed that the ChatGPT sometimes replies with a choice in English (A for example), even though the question was an Arabic question with Arabic letters as choices (أ for instance). In rare cases, the ChatGPT returns that no solution was

---

**Algorithm 1** Extracting the answer chosen by an LLM

```
function PARSELLMANSWER(text, C)
    possible_answers = []
    text = remove_punctuation(text)
    tokens = tokenize(text)
    for c ∈ C do
        if c ∈ tokens then
            possible_answers.append(c)
        end if
    end for
    if |possible_answers| = 1 then
        return possible_answers[0]
    else
        return manual_check(text, possible_answers)
    end if
end function
```

---

found, or that all choices are incorrect. In this case, we assign the LLM's answer to be 'X' (a special letter we use to indicate that the LLM choice was incorrect and not one of the four choices: A, B, C, or D).

One of the popular applications of classical word embedding techniques (*Pennington, Socher & Manning, 2014*; *Bojanowski et al., 2017*; *Mikolov et al., 2013b*; *Mikolov et al., 2013a*) is how they beautifully capture word analogies (*Mikolov et al., 2013a*; *Mikolov, Yih & Zweig, 2013*). We decided to see how well traditional embedding approaches will perform on the GAT word analogies questions. To check for that we used a fastText (*Bojanowski et al., 2017*) model to check which choice amongst the four given choices for a word analogy question is the correct one. The way we determine which choice to pick is highlighted in Algorithm 2. We use the pre-trained word vectors trained on Common Crawl and Wikipedia by fastText (*Grave et al., 2018*) in both Arabic and English analogy questions.

## RESULTS AND DISCUSSION

After we get all the results, we calculate the accuracy of every question type in both languages to measure the Arabic and English linguistic capabilities of ChatGPT. As we can see in Table 2, ChatGPT performance in English is substantially better than Arabic with ChatGPT performing better in English than Arabic by over 28% on average. On the sentence completion task, the difference between ChatGPT's performance in the two languages is staggering with English being superior by a wide margin: 82.67% of English sentence completion questions are answered correctly compared to only 35.85% for Arabic. In all tasks except reading comprehension, ChatGPT fails to answer 40% of the Arabic questions correctly. Overall, ChatGPT was only about to answer 42.74% of the Arabic questions compared to 71.49% of the English questions.

**Algorithm 2** Selecting a pair of words from a provided list (*choices*) that exhibit a comparable relationship to the connection between two given words ($w_1$, $w_2$) using a word embeddings model (*model*)

**function** SELECTBESTPAIR($w_1$, $w_2$, *choices*, *model*)

    $max = -\infty$

    *correct_choice* = *null*

    $v_1 = model[w_1] - model[w_2]$

    **for** $(c_1, c_2) \in$ *choices* **do**

        $v_2 = model[c_1] - model[c_2]$

        $closeness = dot\_product(v_1, v_2)$

        **if** *closeness* > *max* **then**

            $max = closeness$

            *correct_choice* = $(c_1, c_2)$

        **end if**

    **end for**

    **return** *correct_choice*

**end function**

**Table 2** **The accuracy of ChatGPT on the GAT Arabic and English questions.** The best performance is indicated in bold.

| Question type | English | Arabic |
| --- | --- | --- |
| Reading comprehension | **80.22%** | 55.71% |
| Analogy | **54.03%** | 37.19% |
| Contextual error | **68.13%** | 38.61% |
| Sentence completion | **82.67%** | 35.85% |
| **Micro-average** | **71.49%** | 42.74% |
| **Macro-average** | **71.26%** | 41.84% |

It is clear from Table 2 that ChatGPT's performance is better in English than in Arabic. That led us to ask ourselves if we can improve ChatGPT's Arabic performance by using English instructions when asking the model to answer the Arabic questions (and vice-versa for English) as shown in Fig. 4. Table 3 shows the accuracy of ChatGPT on the GAT questions when swapping the instructions. While we see a slight improvement in Arabic when using English task descriptions, on average the effect is not significant. On the sentence completion task, we notice the highest improvement, with English instructions improving the accuracy by 9.43% (from 35.85% to 45.28%). On the other hand, replacing English instructions with Arabic ones degrades performance in all tasks as shown in Table 3. *Lai et al. (2023)* have shown that using English task descriptions always improves performance in Arabic by 8.91% on average across five different tasks. While our results also indicate that English text descriptions lead to an improvement in Arabic's performance on average, the improvement is very small and not significant (< 1.5% improvement on average) and in some cases, it actually degrades the performance. One possible explanation is that *Lai et al. (2023)* machine translated (using Google Translate) their English descriptions which

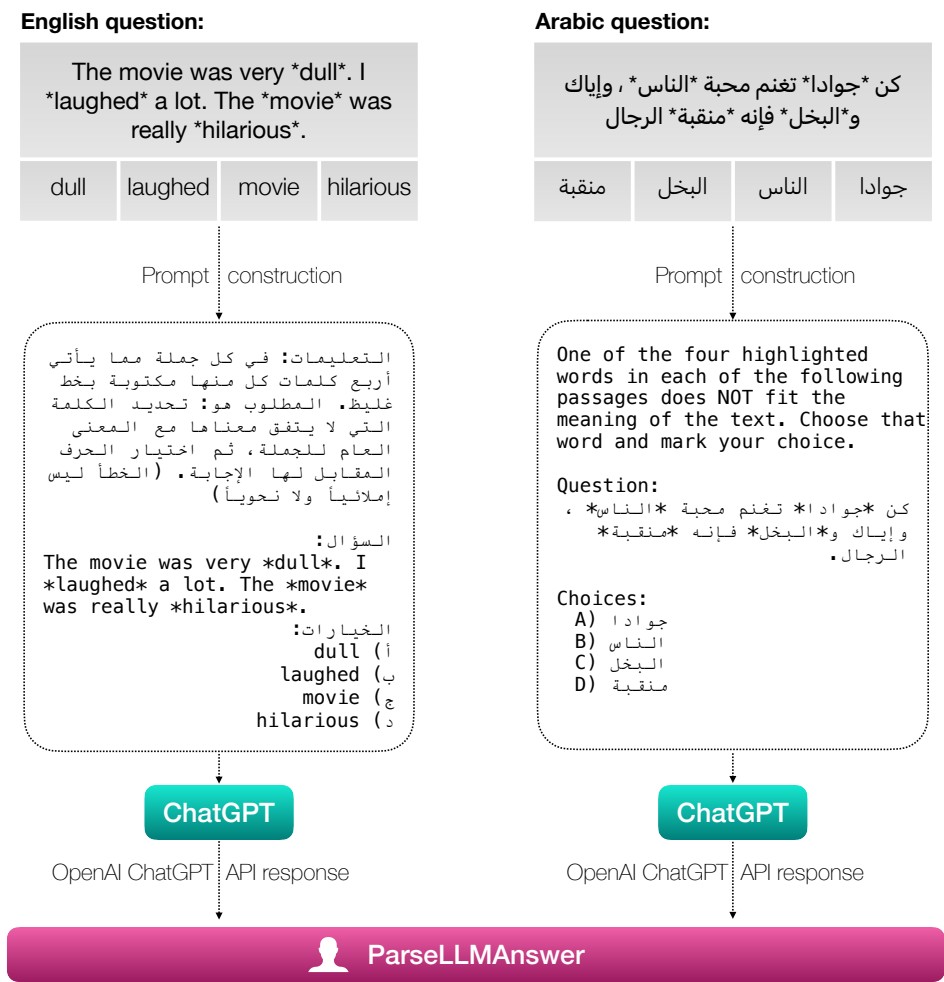

**Figure 4** Swapping Arabic and English task descriptions with each other.

makes the generated non-English descriptions susceptible to inaccuracies in conveying the original description meaning due to the machine translation process. In contrast, we do not translate the task descriptions but use the official ones used in the GAT exam.

Table 4 shows the results of fastText's English and Arabic pre-trained models performance on the English and Arabic GAT analogy questions respectively. It comes surprising that fastText performs better than ChatGPT while being a much smaller and simpler model. It is interesting to note that fastText achieves worse results in English than in Arabic. This is the only instance in all our experiments where Arabic has performed better than English on the same model. One possible explanation for why fastText performed well in Arabic is due to it being subword-based instead of word-based which suits the highly inflected nature of Arabic. *Arora et al. (2020)* showed that contextual Transformer-based models always perform better than classical static word embeddings and they emphasized that this effect is stronger when the text contains a complex internal structure. Our results

**Table 3** **The accuracy of ChatGPT on GAT Arabic questions with English and Arabic instructions and on GAT English questions with English and Arabic instructions.** Arabic$_{AR}$ refers to using Arabic instructions with Arabic questions (Arabic default). Arabic$_{EN}$ refers to using English instructions with Arabic questions. English$_{EN}$ refers to using English instructions with English questions (English default). English$_{AR}$ refers to using Arabic instructions with English questions. The best performance, for both Arabic and English, is indicated in bold.

| Question type | Arabic$_{EN}$ | Arabic$_{AR}$ | English$_{EN}$ | English$_{AR}$ |
|---|---|---|---|---|
| Reading comprehension | 52.86% | **55.71%** | **80.22%** | 75.82% |
| Analogy | **39.67%** | 37.19% | **54.03%** | 51.61% |
| Contextual error | 34.65% | **38.61%** | **68.13%** | 62.64% |
| Sentence completion | **45.28%** | 35.85% | **82.67%** | 77.33% |
| **Micro-average** | **43.80%** | 42.74% | **71.49%** | 67.10% |
| **Macro-average** | **43.12%** | 41.84% | **71.26%** | 66.85% |

**Table 4** **The accuracy of ChatGPT on the GAT Arabic and English analogy questions compared to using fastText embeddings.** The best performance is indicated in bold.

| Language | ChatGPT | fastText |
|---|---|---|
| Arabic | 37.19% | **43.80%** |
| English | **54.03%** | 33.87% |

here shows two different behaviours: (1) static word models can surpass contextual ones in Arabic, and (2) having a complex internal structure doesn't necessarily mean that contextual models would perform better. Arabic has a more complex internal structure than English. Yet, static embeddings in Arabic performs much better than in English as shown in Table 4.

We initially wanted to evaluate GPT-4 (*OpenAI, 2023*), being the most advanced and developed LLM available today, in addition to ChatGPT. However, we do not have access to GPT-4's API, and we do not know when OpenAI will get our application approved. Fortunately, we were able to get (limited) access to GPT-4 through Poe (https://poe.com/) (Platform for Open Exploration) which is a service provided by Quora that allows you to interact with LLMs. They offer GPT-4 access (with a limited number of GPT-4 messages allowed per month) for their paid subscribers. The prompting approach we use with GPT-4 is similar to our original approach highlighted in Fig. 3 with one important difference: we group multiple questions together in one message instead of the one-question-per-request approach that we use with ChatGPT. The reason we do that is due to (1) the reliance on manual copying-and-pasting is slow and cumbersome; it would take a long time to prompt each question separately, and, more importantly, (2) there is a monthly cap on the number of messages we have access to which would've taken us a couple of months to just pass each question in our dataset individually to GPT-4. It is important to note that we only group questions of the same type together as it is done in the GAT exam. We follow the same structure used in the official GAT exam when constructing the prompt for each group: at the top, we have the task description written once (the task description is not repeated) followed by the questions:

```
<question instructions>
```

**Table 5   The accuracy of GPT-4 on the GAT Arabic and English questions compared to ChatGPT.** AR and EN refer to Arabic and English respectively. The best performance is indicated in bold.

| Question type | EN (GPT-4) | AR (GPT-4) | EN (ChatGPT) | AR (ChatGPT) |
|---|---|---|---|---|
| Reading comprehension | **86.81%** | 74.29% | 80.22% | 55.71% |
| Analogy | **73.39%** | 57.02% | 54.03% | 37.19% |
| Contextual error | **83.52%** | 63.37% | 68.13% | 38.61% |
| Sentence completion | **87.33%** | 75.47% | 82.67% | 35.85% |
| **Micro-average** | **82.68%** | 67.74% | 71.49% | 42.74% |
| **Macro-average** | **82.76%** | 67.54% | 71.26% | 41.84% |

```
<question 1>
<question 1 choices>
<question 2>
<question 2 choices>
.
.
.
<question n>
<question n choices>
```

In Table 5 we see the performance of GPT-4 on the GAT dataset. As with ChatGPT, GPT-4 performs better in English than Arabic in all tasks. While GPT-4 performance in both Arabic and English is better than ChatGPT, its performance in Arabic is more impressive jumping from 41.84% to 67.54% ($> 25\%$ improvement). In English, we see the biggest change is in analogy questions where GPT-4 achieves 73.39% compared to 54.03% for ChatGPT; and in Arabic, the largest difference is in sentence completion questions where GPT-4 reaches 75.47% compared to 35.85% for ChatGPT. When comparing the Arabic performance of GPT-4 to the English performance of ChatGPT, we notice that they achieve similar results (67.54% *vs.* 71.26%).

OpenAI mention in their GPT-4 technical report (*OpenAI, 2023*) the performance (accuracy) of ChatGPT and GPT-4 on the MMLU dataset (*Hendrycks et al., 2021b*; *Hendrycks et al., 2021a*) where GPT-4 and ChatGPT achieve 85.5% and 70.1% respectively. These numbers are similar to the results we get in our benchmark for GPT-4 and ChatGPT (82.76% and 71.26% respectively) as shown in Table 6. What is interesting is OpenAI's claim that GPT-4 multilingual performance surpasses the English language performance of ChatGPT for many languages including Arabic. To reach this conclusion they translated the MMLU dataset to Arabic using Microsoft's Azure Translate and then measured GPT-4's accuracy on the Arabic machine translated dataset where it achieves an impressive 80.0%. In the appendix of their paper, they mention that the machine translation process can sometimes keep proper nouns in English which can result in an improved performance. While our results show that GPT-4 gains in Arabic are more than it's gains in English

**Table 6 The accuracy of ChatGPT and GPT-4 on the MMLU and GAT benchmarks.** MMLU results are taken from *OpenAI (2023)*. Δ refers to the difference between MMLU and GAT. AR and EN refer to Arabic and English respectively.

|  | MMLU | GAT | Δ |
|---|---|---|---|
| EN (ChatGPT) | 70.10% | 71.25% | −1.15% |
| EN (GPT-4) | 85.50% | 82.76% | 2.74% |
| AR (GPT-4) | 80.00% | 67.54% | 12.46% |

compared to ChatGPT, it doesn't indicate that the Arabic performance of GPT-4 (67.54%) is better than ChatGPT's English performance (71.26%) contradicting OpenAI's claim.

We notice a current trend in NLP research today where researchers rely on machine translated content to evaluate LLMs (*OpenAI, 2023*; *Lai et al., 2023*). The machine translation process is prone to inaccuracies in faithfully conveying the intended meaning of the original description, and introduces a susceptibility that can lead to deviations and misunderstandings. Our results seem to contradict two previous reported results: (1) Replacing Arabic task descriptions with English ones improves performance (*Lai et al., 2023*), and (2) GPT-4 performance in Arabic is better than ChatGPT's performance in English (*OpenAI, 2023*). In both these papers, the authors used machine translated text instead of relying on a benchmark that is carefully designed for Arabic.

## Limitations

There are two limitations in the work we have done that we would like to discuss. The first one is that we cannot guarantee that ChatGPT (or GPT-4) has not been trained on the data (GAT questions). Nonetheless, we believe that they have not seen all the data during their pre-training stage since substantial parts of the data we collected required logging in to the NCA website, extracting text from images, or dealing with PDFs. Moreover, the answers for GAT questions are not usually presented with the question itself and require looking for an answer key located on a different page, clicking on a link, submitting a form, or other extra steps to reach them. The second one relates to prompting: the prompting approaches we use differ between ChatGPT and GPT-4 due to the limitations and constraints of the GPT-4 access we had. As it is widely known, LLMs are sensitive to prompting, and changing the prompts used can improve/degrade performance. To mitigate this effect, in both our prompts (for ChatGPT and GPT-4) we always follow the task descriptions found in the official GAT guide.

## CONCLUSION

This research article presents a pioneering benchmark for the bilingual evaluation of LLMs in English and Arabic, addressing the existing limitations in Arabic LLM assessment. By leveraging the GAT as a benchmark, we provide a robust and standardized method for comparing LLM performance across the two languages. Our experiments reveal significant differences in the linguistic abilities of ChatGPT in English and Arabic and demonstrate the impact of task description language-switching. Interestingly, we observe that fastText, a traditional word embedding model, outperforms ChatGPT in word

analogies. Most notably, our findings highlight the superior Arabic linguistic capabilities of GPT-4 compared to ChatGPT and suggest that they are almost on par with ChatGPT's English performance. This work paves the way for future research and improvement of LLMs in Arabic and, more importantly, other non-English languages.

### Funding
The authors received no funding for this work.

### Competing Interests
The author declares there are no competing interests.

### Author Contributions
- Mohamed Alkaoud conceived and designed the experiments, performed the experiments, analyzed the data, performed the computation work, prepared figures and/or tables, authored or reviewed drafts of the article, and approved the final draft.

### Data Availability
The GAT English and Arabic questions are available in the Supplementary Files.

### Supplemental Information
Supplemental information for this article can be found online at http://dx.doi.org/10.7717/peerj-cs.1893#supplemental-information.

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
