# Peer review of "A bilingual benchmark for evaluating large language models"

_PeerJ Computer Science, doi:10.7717/peerj-cs.1893_

## Round 0.1 · original submission · Major Revisions

Please respond to the reviewers' comments, especially those from Reviewer 2.

Reviewer 1 ·

Basic reporting

This paper compares the skill of chatGPT and GPT4 in answering GAT exam questions in Arabic and English. The idea is to provide a benchmark data that can be used to evaluate the skill of the models with respect to the same task but formulated in different languages. Samples of GAT exam questions in Arabic and English were fed to the two models and performance was reported and discussed.

The paper is well-written and easy to follow.

Experimental design

The experiments in this paper are explained nicely.

Validity of the findings

The findings are valid and reasonable and align with the findings of the research community working on the specific topic addressed by this research.

Additional comments

I have the following remarks for the authors:
1. Algorithm1: it is not clear how to extract the correct answer which is returned by ChatGPT. Simply checking if the choice (c) belongs to the tokens of the question is not enough. I guess you need to check that algorithm again.
2. 456 questions in English and 468 in Arabic is small to be called benchmark for testing language models. Also, the paper did not describe the details and nature of crowdsourcing which was used to generate this dataset.
3. There are 14 references taken from arXiv. Kindly check if these are published elsewhere and modify these references accordingly. Papers in arXiv are not peer reviewed.
4. References no 44, 45 and 52 lack information. Try to provide all information related to these references.
5. Paragraph 4 in the Introduction Section: add references to GAT, NCA and KFPUM.
6. Introduction section: add organization of the paper at the end of this Section
7. Line 200 page 4: “with four possible answers (A, B, C, D)” These are not possible answers, rather they are given choices and there is only one correct answer. I guess this sentence needs paraphrasing.
8. Equation 1: I believe you should use MAX rather than MIN because you are looking for the most similar pairs.
9. Page 8, line 272: the text in this line seems cutoff.

Cite this review as

Reviewer 2 ·

Basic reporting

The paper presents a new benchmark dataset for multilingual large language models in Arabic and English. The dataset was collected using GAT exams verbal section. The paper primarily focuses on detailing the data collection process, with no substantial technical contributions. The paper also has various limitations, which are outlined as follows.

• The introduction lacks a clear structure of introducing the problem and providing summary of effort in the field and gap in the research area.
• In the introduction, the author stated that one of the contributions of the paper is to propose a new way of evaluating LLMs in Arabic and English. It's unclear how the author introduced a novel approach to evaluating LLMS. While data collection is crucial, it doesn't necessarily imply that the author has proposed a new method for assessing LLMS.
• The background section appears extensive and may not be necessary since it mainly describes other proposed approaches (transformer-based methods) not sure how these methods are relevant to this paper.

Experimental design

• The author didn’t provide enough details of the methodology or the results. For example, in the approach section which supposed to provide details of the methodology, there is no description of how fast text was implemented.

• Equation 1 , should be argmax since the function should return the part with maximum similarity.

Validity of the findings

• The author stated that no previous dataset existed for Arabic LLM. However, as demonstrated in the following study, a dataset has been created for this specific purpose.

Ali, Abbas Raza, et al. "A Large and Diverse Arabic Corpus for Language Modeling." arXiv preprint arXiv:2201.09227 (2022).

• The findings are addressed but not reflected in terms of new insights gained in comparison to previous studies.

Cite this review as

---

## Round 0.2 · accepted · Accept

Reviewer 2 considered his/her comments have been addressed.

Reviewer 2 ·

Basic reporting

no comment

Experimental design

no comment

Validity of the findings

no comment

Additional comments

Reviewer comments addressed.

Cite this review as